# Boundary-Seeking Generative Adversarial Networks

**R Devon Hjelm**[*]
MILA, University of Montréal, IVADO
erroneus@gmail.com

**Athul Paul Jacob**[*]
MILA, MSR, University of Waterloo
apjacob@edu.uwaterloo.ca

**Tong Che**
MILA, University of Montréal
tong.che@umontreal.ca

**Adam Trischler**
MSR
adam.trischler@microsoft.com

**Kyunghyun Cho**
New York University,
CIFAR Azrieli Global Scholar
kyunghyun.cho@nyu.edu

**Yoshua Bengio**
MILA, University of Montréal, CIFAR, IVADO
yoshua.bengio@umontreal.ca

## Abstract

Generative adversarial networks (GANs, Goodfellow et al., 2014) are a learning framework that rely on training a discriminator to estimate a measure of difference between a target and generated distributions. GANs, as normally formulated, rely on the generated samples being completely differentiable w.r.t. the generative parameters, and thus do not work for discrete data. We introduce a method for training GANs with discrete data that uses the estimated difference measure from the discriminator to compute importance weights for generated samples, thus providing a policy gradient for training the generator. The importance weights have a strong connection to the decision boundary of the discriminator, and we call our method *boundary-seeking GANs* (BGANs). We demonstrate the effectiveness of the proposed algorithm with discrete image and character-based natural language generation. In addition, the boundary-seeking objective extends to continuous data, which can be used to improve stability of training, and we demonstrate this on Celeba, Large-scale Scene Understanding (LSUN) bedrooms, and Imagenet without conditioning.

## 1 Introduction

Generative adversarial networks (GAN, Goodfellow et al., 2014) involve a unique generative learning framework that uses two separate models, a generator and discriminator, with opposing or *adversarial* objectives. Training a GAN only requires back-propagating a learning signal that originates from a learned objective function, which corresponds to the loss of the discriminator trained in an adversarial manner. This framework is powerful because it trains a generator without relying on an explicit formulation of the probability density, using only samples from the generator to train.

GANs have been shown to generate often-diverse and realistic samples even when trained on high-dimensional large-scale continuous data (Radford et al., 2015). GANs however have a serious limitation on the type of variables they can model, because they require the composition of the generator and discriminator to be fully differentiable.

With discrete variables, this is not true. For instance, consider using a step function at the end of a generator in order to generate a discrete value. In this case, back-propagation alone cannot provide the training signal, because the derivative of a step function is 0 almost everywhere. This is problematic, as many important real-world datasets are discrete, such as character- or word-based

---

[*]Denotes first-author contributions.

representations of language. The general issue of credit assignment for computational graphs with discrete operations (e.g. discrete stochastic neurons) is difficult and open problem, and only approximate solutions have been proposed in the past (Bengio et al., 2013; Gu et al., 2015; Gumbel & Lieblein, 1954; Jang et al., 2016; Maddison et al., 2016; Tucker et al., 2017). However, none of these have yet been shown to work with GANs. In this work, we make the following contributions:

- We provide a theoretical foundation for *boundary-seeking GANs* (BGAN), a principled method for training a generator of discrete data using a discriminator optimized to estimate an $f$-divergence (Nguyen et al., 2010; Nowozin et al., 2016). The discriminator can then be used to formulate *importance weights* which provide policy gradients for the generator.
- We verify this approach quantitatively works across a set of $f$-divergences on a simple classification task and on a variety of image and natural language benchmarks.
- We demonstrate that BGAN performs quantitatively better than WGAN-GP (Gulrajani et al., 2017) in the simple discrete setting.
- We show that the boundary-seeking objective extends theoretically to the continuous case and verify it works well with some common and difficult image benchmarks. Finally, we show that this objective has some improved stability properties within training and without.

## 2 BOUNDARY-SEEKING GANS

In this section, we will introduce boundary-seeking GANs (BGAN), an approach for training a generative model adversarially with discrete data, as well as provide its theoretical foundation. For BGAN, we assume the normal generative adversarial learning setting commonly found in work on GANs (Goodfellow et al., 2014), but these ideas should extend elsewhere.

### 2.1 GENERATIVE ADVERSARIAL LEARNING AND PROBLEM STATEMENT

Assume that we are given empirical samples from a target distribution, $\{x^{(i)} \in \mathcal{X}\}_{i=1}^M$, where $\mathcal{X}$ is the domain (such as the space of images, word- or character- based representations of natural language, etc.). Given a random variable $Z$ over a space $\mathcal{Z}$ (such as $[0,1]^m$), we wish to find the optimal parameters, $\hat{\theta} \in \mathbb{R}^d$, of a function, $G_\theta : \mathcal{Z} \to \mathcal{X}$ (such as a deep neural network), whose induced probability distribution, $\mathbb{Q}_\theta$, describes well the empirical samples.

In order to put this more succinctly, it is beneficial to talk about a probability distribution of the empirical samples, $\mathbb{P}$, that is defined on the same space as $\mathbb{Q}_\theta$. We can now consider the *difference measure* between $\mathbb{P}$ and $\mathbb{Q}_\theta$, $D(\mathbb{P}, \mathbb{Q}_\theta)$, so the problem can be formulated as finding the parameters:

$$\hat{\theta} = \arg\min_\theta D(\mathbb{P}, \mathbb{Q}_\theta). \tag{1}$$

Defining an appropriate difference measure is a long-running problem in machine learning and statistics, and choosing the best one depends on the specific setting. Here, we wish to avoid making strong assumptions on the exact forms of $\mathbb{P}$ or $\mathbb{Q}_\theta$, and we desire a solution that is scalable and works with very high dimensional data. Generative adversarial networks (GANs, Goodfellow et al., 2014) fulfill these criteria by introducing a *discriminator function*, $D_\phi : \mathcal{X} \to \mathbb{R}$, with parameters, $\phi$, then defining a *value function*,

$$\mathcal{V}(\mathbb{P}, \mathbb{Q}_\theta, D_\phi) = \mathbb{E}_\mathbb{P}\left[\log D_\phi(x)\right] + \mathbb{E}_{h(z)}\left[\log(1 - D_\phi(G(z)))\right], \tag{2}$$

where samples $z$ are drawn from a simple prior, $h(z)$ (such as $U(0,1)$ or $\mathcal{N}(0,1)$). Here, $D_\phi$ is a neural network with a sigmoid output activation, and as such can be interpreted as a simple binary classifier, and the value function can be interpreted as the negative of the *Bayes risk*. GANs train the discriminator to maximize this value function (minimize the mis-classification rate of samples coming from $\mathbb{P}$ or $\mathbb{Q}_\theta$), while the generator is trained to minimize it. In other words, GANs solve an optimization problem:

$$(\hat{\theta}, \hat{\phi}) = \arg\min_\theta \arg\max_\phi \mathcal{V}(\mathbb{P}, \mathbb{Q}_\theta, D_\phi). \tag{3}$$

Optimization using only back-propogation and stochastic gradient descent is possible when the generated samples are completely differentiable w.r.t. the parameters of the generator, $\theta$.

In the non-parametric limit of an optimal discriminator, the value function is equal to a scaled and shifted version of the Jensen-Shannon divergence, $2 * \mathcal{D}_{JSD}(\mathbb{P}||\mathbb{Q}_\theta) - \log 4$,[1] which implies the generator is minimizing this divergence in this limit. $f$-GAN (Nowozin et al., 2016) generalized this idea over all $f$-divergences, which includes the Jensen-Shannon (and hence also GANs) but also the Kullback–Leibler, Pearson $\chi^2$, and squared-Hellinger. Their work provides a nice formalism for talking about GANs that use $f$-divergences, which we rely on here.

**Definition 2.1** ($f$-divergence and its dual formulation). Let $f : \mathbb{R}_+ \to \mathbb{R}$ be a convex lower semi-continuous function and $f^\star : \mathcal{C} \subseteq \mathbb{R} \to \mathbb{R}$ be the convex conjugate with domain $\mathcal{C}$. Next, let $\mathcal{T}$ be an arbitrary family of functions, $\mathcal{T} = \{T : \mathcal{X} \to \mathcal{C}\}$. Finally, let $\mathbb{P}$ and $\mathbb{Q}$ be distributions that are completely differentiable w.r.t. the same Lebesgue measure, $\mu$.[2] The $f$-divergence, $\mathcal{D}_f(\mathbb{P}||\mathbb{Q}_\theta)$, generated by $f$, is bounded from below by its dual representation (Nguyen et al., 2010),

$$\mathcal{D}_f(\mathbb{P}||\mathbb{Q}) = \mathbb{E}_{\mathbb{Q}}\left[f\left(\frac{d\mathbb{P}/d\mu}{d\mathbb{Q}/d\mu}\right)\right] \geq \sup_{T \in \mathcal{T}}\left(\mathbb{E}_{\mathbb{P}}[T(x)] - \mathbb{E}_{\mathbb{Q}}[f^\star(T(x))]\right). \tag{4}$$

The inequality becomes *tight* when $\mathcal{T}$ is the family of all possible functions. The dual form allows us to change a problem involving likelihood ratios (which may be intractable) to an maximization problem over $\mathcal{T}$. This sort of optimization is well-studied if $\mathcal{T}$ is a family of neural networks with parameters $\phi$ (a.k.a., *deep learning*), so the supremum can be found with gradient ascent (Nowozin et al., 2016).

**Definition 2.2** (Variational lower-bound for the $f$-divergence). Let $T_\phi = \nu \circ F_\phi$ be a function, which is the composition of an activation function, $\nu : \mathbb{R} \to \mathcal{C}$ and a neural network, $F_\phi : \mathcal{X} \to \mathbb{R}$. We can write the *variational lower-bound* of the supremum in Equation 4 as [3]:

$$\mathcal{D}_f(\mathbb{P}||\mathbb{Q}_\theta) \geq \mathbb{E}_{\mathbb{P}}[\nu \circ F_\phi(x)] - \mathbb{E}_{\mathbb{Q}_\theta}[f^\star(\nu \circ F_\phi(x))] = \mathcal{V}(\mathbb{P}, \mathbb{Q}_\theta, T_\phi). \tag{5}$$

Maximizing Equation 5 provides a neural estimator of $f$-divergence, or *neural divergence* (Huang et al., 2018). Given the family of neural networks, $\mathcal{T}_\Phi = \{T_\phi\}_{\phi \in \Phi}$, is sufficiently expressive, this bound can become arbitrarily tight, and the neural divergence becomes arbitrarily close to the true divergence. As such, GANs are extremely powerful for training a generator of continuous data, leveraging a dual representation along with a neural network with theoretically unlimited capacity to estimate a difference measure.

For the remainder of this work, we will refer to $T_\phi = \nu \circ F_\phi$ as the *discriminator* and $F_\phi$ as the *statistic network* (which is a slight deviation from other works). We use the general term *GAN* to refer to all models that simultaneously minimize and maximize a *variational lower-bound*, $\mathcal{V}(\mathbb{P}, \mathbb{Q}_\theta, T_\phi)$, of a difference measure (such as a divergence or distance). In principle, this extends to variants of GANs which are based on integral probability metrics (IPMs, Sriperumbudur et al., 2009) that leverage a dual representation, such as those that rely on restricting $\mathcal{T}$ through parameteric regularization (Arjovsky et al., 2017) or by constraining its output distribution (Mroueh & Sercu, 2017; Mroueh et al., 2017; Sutherland et al., 2016).

## 2.2 Estimation of the target distribution

Here we will show that, with the variational lower-bound of an $f$-divergence along with a family of positive activation functions, $\nu : \mathbb{R} \to \mathbb{R}_+$, we can estimate the target distribution, $\mathbb{P}$, using the generated distribution, $\mathbb{Q}_\theta$, and the discriminator, $T_\phi$.

**Theorem 1.** *Let $f$ be a convex function and $T^\star \in \mathcal{T}$ a function that satisfies the supremum in Equation 4 in the non-parametric limit. Let us assume that $\mathbb{P}$ and $\mathbb{Q}_\theta(x)$ are absolutely continuous w.r.t. a measure $\mu$ and hence admit densities, $p(x)$ and $q_\theta(x)$. Then the target density function, $p(x)$, is equal to $(\partial f^\star/\partial T)(T^\star(x))q_\theta(x)$.*

---

[1]Note that this has an absolute minimum, so that the above optimization is a Nash-equilibrium

[2]$\mu$ can be thought of in this context as $x$, so that it can be said that $\mathbb{P}$ and $\mathbb{Q}$ have density functions on $x$.

[3]It can be easily verified that, for $\nu(y) = -\log(1 + e^{-y})$, $f(u) = u \log u + (1 + u) \log(1 + u)$, and setting $T = \log D$, the variational lower-bound becomes exactly equal to the GAN value function.

Table 1: Important weights and nonlinearities that ensure

| Importance weights for $f$-divergences | | |
|---|---|---|
| $f$-divergence | $\nu(y)$ | $w(x) = (\partial f^\star / \partial T)(T(x))$ |
| GAN | $-\log\left(1 + e^{-y}\right)$ | $-\frac{1}{1-e^{-T_\phi}} = e^{F_\phi(x)}$ |
| Jensen-Shannon | $\log 2 - \log\left(1 + e^{-y}\right)$ | $-\frac{1}{2-e^{-T_\phi}} = e^{F_\phi(x)}$ |
| KL | $y + 1$ | $e^{(T_\phi(x)-1)} = e^{F_\phi(x)}$ |
| Reverse KL | $-e^{-y}$ | $-\frac{1}{T_\phi(x)} = e^{F_\phi(x)}$ |
| Squared-Hellinger | $1 - e^{-v/2}$ | $\frac{1}{(1-T_\phi(x))^2} = e^{F_\phi(x)}$ |

*Proof.* Following the definition of the $f$-divergence and the convex conjugate, we have:

$$\mathcal{D}_f(\mathbb{P}||\mathbb{Q}_\theta) = \mathbb{E}_{\mathbb{Q}_\theta}\left[f\left(\frac{p(x)}{q(x)}\right)\right] = \mathbb{E}_{\mathbb{Q}_\theta}\left[\sup_t\left\{t\frac{p(x)}{q(x)} - f^\star(t)\right\}\right]. \tag{6}$$

As $f^\star$ is convex, there is an absolute maximum when $\frac{\partial f^\star}{\partial t}(t) = \frac{p(x)}{q_\theta(x)}$. Rephrasing $t$ as a function, $T(x)$, and by the definition of $T^\star(x)$, we arrive at the desired result. $\square$

Theorem 1 indicates that the target density function can be re-written in terms of a generated density function and a scaling factor. We refer to this scaling factor, $w^\star(x) = (\partial f^\star / \partial T)(T^\star(x))$, as the optimal *importance weight* to make the connection to importance sampling [4]. In general, an optimal discriminator is hard to guarantee in the saddle-point optimization process, so in practice, $T_\phi$ will define a lower-bound that is not exactly tight w.r.t. the $f$-divergence. Nonetheless, we can define an estimator for the target density function using a sub-optimal $T_\phi$.

**Definition 2.3** ($f$-divergence importance weight estimator)**.** Let $f$ and $f^\star$, and $T_\phi(x)$ be defined as in Definitions 2.1 and 2.2 but where $\nu : \mathbb{R} \to \mathbb{R}_+ \subseteq \mathcal{C}$ is a positive activation function. Let $w(x) = (\partial f^\star / \partial T)(T(x))$ and $\beta = \mathbb{E}_{\mathbb{Q}_\phi}[w(x)]$ be a partition function. The $f$-divergence importance weight estimator, $\tilde{p}(x)$ is

$$\tilde{p}(x) = \frac{w(x)}{\beta} q_\theta(x). \tag{7}$$

The non-negativity of $\nu$ is important as the densities are positive. Table 1 provides a set of $f$-divergences (following suggestions of Nowozin et al. (2016) with only slight modifications) which are suitable candidates and yield positive importance weights. Surprisingly, each of these yield the same function over the neural network before the activation function: $w(x) = e^{F_\phi(x)}$.[5] It should be noted that $\tilde{p}(x)$ is a potentially biased estimator for the true density; however, the bias only depends on the tightness of the variational lower-bound: the tighter the bound, the lower the bias. This problem reiterates the problem with all GANs, where proofs of convergence are only provided in the optimal or near-optimal limit (Goodfellow et al., 2014; Nowozin et al., 2016; Mao et al., 2016).

### 2.3 BOUNDARY-SEEKING GANS

As mentioned above and repeated here, GANs only work *when the value function is completely differentiable w.r.t. the parameters of the generator, $\theta$.* The gradients that would otherwise be used to train the generator of discrete variables are zero almost everywhere, so it is impossible to train the generator directly using the value function. Approximations for the back-propagated signal exist (Bengio et al., 2013; Gu et al., 2015; Gumbel & Lieblein, 1954; Jang et al., 2016; Maddison et al., 2016; Tucker et al., 2017), but as of this writing, none has been shown to work satisfactorily in training GANs with discrete data.

Here, we introduce the boundary-seeking GAN as a method for training GANs with discrete data. We first introduce a policy gradient based on the KL-divergence which uses the importance weights

---

[4] In the case of the $f$-divergence used in Goodfellow et al. (2014), the optimal importance weight equals $w^\star(x) = e^{F^\star(x)} = D^\star(x)/(1 - D^\star(x))$

[5] Note also that the normalized weights resemble softmax probabilities

---

**Algorithm 1** . Discrete Boundary Seeking GANs

$(\theta, \phi) \leftarrow$ initialize the parameters of the generator and statistic network
**repeat**
    $\hat{x}^{(n)} \sim \mathbb{P}$                     $\triangleright$ Draw $N$ samples from the empirical distribution
    $z^{(n)} \sim h(z)$                       $\triangleright$ Draw $N$ samples from the prior distribution
    $x^{(m|n)} \sim g_\theta(x \mid z^{(n)})$        $\triangleright$ Draw $M$ samples from each conditional $g_\theta(x \mid z^{(m)})$ (drawn
independently if $\mathbb{P}$ and $\mathbb{Q}_\theta$ are multi-variate)
    $w(x^{(m|n)}) \leftarrow (\partial f^\star / \partial T) \circ (\nu \circ F_\phi(x^{(m|n)}))$
    $\tilde{w}(x^{(m|n)}) \leftarrow w(x^{(m|n)}) / \sum_{m'} w(x^{(m'|n)})$      $\triangleright$ Compute the un-normalized and normalized
importance weights (applied uniformly if $\mathbb{P}$ and $\mathbb{Q}_\theta$ are multi-variate)
    $\mathcal{V}(\mathbb{P}, \mathbb{Q}_\theta, T_\phi) \leftarrow \frac{1}{N} \sum_n F_\phi(\hat{x}^{(n)}) - \frac{1}{N} \sum_n \frac{1}{M} \sum_m w(x^{(m|n)})$      $\triangleright$ Estimate the variational
lower-bound
    $\phi \leftarrow \phi + \gamma_d \nabla_\phi \mathcal{V}(\mathbb{P}, \mathbb{Q}_\theta, T_\phi)$             $\triangleright$ Optimize the discriminator parameters
    $\theta \leftarrow \theta + \gamma_g \frac{1}{N} \sum_{n,m} \tilde{w}(x^{(m|n)}) \nabla_\theta \log g_\theta(x^{(m|n)} \mid z)$     $\triangleright$ Optimize the generator parameters
**until** convergence

---

as a reward signal. We then introduce a lower-variance gradient which defines a unique reward signal for each $z$ and prove this can be used to solve our original problem.

**Policy gradient based on importance sampling** Equation 7 offers an option for training a generator in an adversarial way. If we know the explicit density function, $q_\theta$, (such as a multivariate Bernoulli distribution), then we can, using $\tilde{p}(x)$ as a target (keeping it fixed w.r.t. optimization of $\theta$), train the generator using the gradient of the KL-divergence:

$$\nabla_\theta \mathcal{D}_{KL}(\tilde{p}(x) || q_\theta) = -\mathbb{E}_{\mathbb{Q}_\theta} \left[ \frac{w(x)}{\beta} \nabla_\theta \log q_\theta(x) \right]. \tag{8}$$

Here, the connection to importance sampling is even clearer, and this gradient resembles other importance sampling methods for training generative models in the discrete setting (Bornschein & Bengio, 2014; Rubinstein & Kroese, 2016). However, we expect the variance of this estimator will be high, as it requires estimating the partition function, $\beta$ (for instance, using Monte-Carlo sampling). We address reducing the variance from estimating the normalized importance weights next.

**Lower-variance policy gradient** Let $q_\theta(x) = \int_\mathcal{Z} g_\theta(x \mid z) h(z) dz$ be a probability density function with a conditional density, $g_\theta(x \mid z) : \mathcal{Z} \to [0, 1]^d$ (e.g., a multivariate Bernoulli distribution), and prior over $z$, $h(z)$. Let $\alpha(z) = \mathbb{E}_{g_\theta(x|z)}[w(x)] = \int_\mathcal{X} g_\theta(x \mid z) w(x) dx$ be a partition function over the conditional distribution. Let us define $\tilde{p}(x \mid z) = \frac{w(x)}{\alpha(z)} g_\theta(x \mid z)$ as the (normalized) conditional distribution weighted by $\frac{w(x)}{\alpha(z)}$. The expected conditional KL-divergence over $h(z)$ is:

$$\mathbb{E}_{h(z)}[\mathcal{D}_{KL}(\tilde{p}(x \mid z) || g_\theta(x \mid z))] = \int_\mathcal{Z} h(z) \mathcal{D}_{KL}(\tilde{p}(x \mid z) || g_\theta(x \mid z)) \, dz \tag{9}$$

Let $x^{(m)} \sim g_\theta(x \mid z)$ be samples from the prior and $\tilde{w}(x^{(m)}) = \frac{w(x^{(m)})}{\sum_{m'} w(x^{(m')})}$ be a Monte-Carlo estimate of the normalized importance weights. The gradient of the expected conditional KL-divergence w.r.t. the generator parameters, $\theta$, becomes:

$$\nabla_\theta \mathbb{E}_{h(z)}[\mathcal{D}_{KL}(\tilde{p}(x \mid z) || g_\theta(x \mid z))] = -\mathbb{E}_{h(z)} \left[ \sum_m \tilde{w}(x^{(m)}) \nabla_\theta \log g_\theta(x^{(m)} \mid z) \right], \tag{10}$$

where we have approximated the expectation using the Monte-Carlo estimate.

Minimizing the expected conditional KL-divergences is stricter than minimizing the KL-divergence in Equation 7, as it requires all of the conditional distributions to match independently. We show that the KL-divergence of the marginal probabilities is zero when the expectation of the conditional KL-divergence is zero as well as show this estimator works better in practice in the Appendix.

Algorithm 1 describes the training procedure for discrete BGAN. This algorithm requires an additional $M$ times more computation to compute the normalized importance weights, though these can be computed in parallel exchanging space for time. When the $\mathbb{P}$ and $\mathbb{Q}_\theta$ are multi-variate (such as with discrete image data), we make the assumption that the observed variables are independent conditioned on $Z$. The importance weights, $w$, are then applied *uniformly* across each of the observed variables.

**Connection to policy gradients** REINFORCE is a common technique for dealing with discrete data in GANs (Che et al., 2017; Li et al., 2017). Equation 9 is a policy gradient in the special case that the reward is the normalized importance weights. This reward approaches the likelihood ratio in the non-parametric limit of an optimal discriminator. Here, we make another connection to REINFORCE as it is commonly used, with baselines, by deriving the gradient of the reversed KL-divergence.

**Definition 2.4** (REINFORCE-based BGAN). Let $T_\phi(x)$ be defined as above where $\partial f^\star/\partial T(T_\phi(x)) = e^{F_\phi(x)}$. Consider the gradient of the *reversed* KL-divergence:

$$\nabla_\theta \mathcal{D}_{KL}\left(q_\theta \| \tilde{p}\right) = -\mathbb{E}_{h(z)}\left[\sum_m (\log w(x^{(m)}) - \log \beta + 1)\nabla_\theta \log g_\theta(x^{(m)} \mid z)\right]$$

$$= -\mathbb{E}_{h(z)}\left[\sum_m (F_\phi(x) - b)\nabla_\theta \log g_\theta(x^{(m)} \mid z)\right] \qquad (11)$$

From this, it is clear that we can consider the output of the statistic network, $F_\phi(x)$, to be a *reward* and $b = \log \beta = \mathbb{E}_{\mathbb{Q}_\theta}[w(x)]$ to be the analog of a baseline.[6] This gradient is similar to those used in previous works on discrete GANs, which we discuss in more detail in Section 3.

## 2.4 CONTINUOUS VARIABLES AND THE STABILITY OF GANS

For continuous variables, minimizing the variational lower-bound suffices as an optimization technique as we have the full benefit of back-propagation to train the generator parameters, $\theta$. However, while the convergence of the discriminator is straightforward, to our knowledge there is no general proof of convergence for the generator except in the non-parametric limit or near-optimal case. What's worse is the value function can be arbitrarily large and negative. Let us assume that $\max T = M < \infty$ is unique. As $f^\star$ is convex, the minimum of the lower-bound over $\theta$ is:

$$\inf_\theta \mathcal{V}(\mathbb{P}, \mathbb{Q}_\theta, D_\phi) = \inf_\theta \mathbb{E}_{\mathbb{P}}[T_\phi(x)] - \mathbb{E}_{\mathbb{Q}_\theta}[f^\star(T_\phi(x))]$$

$$= \mathbb{E}_{\mathbb{P}}[T_\phi(x)] - \sup_\theta \mathbb{E}_{\mathbb{Q}_\theta}[f^\star(T_\phi(x))] = \mathbb{E}_{\mathbb{P}}[T_\phi(x)] - f^\star(M). \qquad (12)$$

In other words, the generator objective is optimal when the generated distribution, $\mathbb{Q}_\theta$, is nonzero only for the set $\{x \mid T(x) = M\}$. Even outside this worst-case scenario, the additional consequence of this minimization is that this variational lower-bound can become looser w.r.t. the $f$-divergence, with no guarantee that the generator would actually improve. Generally, this is avoided by training the discriminator in conjunction with the generator, possibly for many steps for every generator update. However, this clearly remains one source of potential instability in GANs.

Equation 7 reveals an alternate objective for the generator that should improve stability. Notably, we observe that for a given estimator, $\tilde{p}(x)$, $q_\theta(x)$ matches when $w(x) = (\partial f^\star/\partial T)(T(x)) = 1$.

**Definition 2.5** (Continuous BGAN objective for the generator). Let $G_\theta : \mathcal{Z} \to \mathcal{X}$ be a generator function that takes as input a latent variable drawn from a simple prior, $z \sim h(z)$. Let $T_\phi$ and $w(x)$ be defined as above. We define the continuous BGAN objective as: $\hat{\theta} = \arg\min_\theta(\log w(G_\theta(z)))^2$. We chose the $\log$, as with our treatments of $f$-divergences in Table 1, the objective is just the square of the statistic network output:

$$\hat{\theta} = \arg\min_\theta F_\phi(G_\theta(z))^2. \qquad (13)$$

This objective can be seen as changing a concave optimization problem (which is poor convergence properties) to a convex one.

---

[6]Note that we have removed the additional constant as $\mathbb{E}_{q_\theta}[1 * \nabla_\theta q_\theta] = 0$

## 3    RELATED WORK AND DISCUSSION

**On estimating likelihood ratios from the discriminator**    Our work relies on estimating the likelihood ratio from the discriminator, the theoretical foundation of which we draw from $f$-GAN (Nowozin et al., 2016). The connection between the likelihood ratios and the policy gradient is known in previous literature (Jie & Abbeel, 2010), and the connection between the discriminator output and the likelihood ratio was also made in the context of continuous GANs (Mohamed & Lakshminarayanan, 2016; Tran et al., 2017). However, our work is the first to successfully formulate and apply this approach to the discrete setting.

**Importance sampling**    Our method is very similar to re-weighted wake-sleep (RWS, Bornschein & Bengio, 2014), which is a method for training Helmholtz machines with discrete variables. RWS also relies on minimizing the KL divergence, the gradients of which also involve a policy gradient over the likelihood ratio. Neural variational inference and learning (NVIL, Mnih & Gregor, 2014), on the other hand, relies on the reverse KL. These two methods are analogous to our importance sampling and REINFORCE-based BGAN formulations above.

**GAN for discrete variables**    Training GANs with discrete data is an active and unsolved area of research, particularly with language model data involving recurrent neural network (RNN) generators (Yu et al., 2016; Li et al., 2017). Many REINFORCE-based methods have been proposed for language modeling (Yu et al., 2016; Li et al., 2017; Dai et al., 2017) which are similar to our REINFORCE-based BGAN formulation and effectively use the sigmoid of the estimated log-likelihood ratio. The primary focus of these works however is on improving *credit assignment*, and their approaches are compatible with the policy gradients provided in our work.

There have also been some improvements recently on training GANs on language data by rephrasing the problem into a GAN over some continuous space (Lamb et al., 2016; Kim et al., 2017; Gulrajani et al., 2017). However, each of these works bypass the difficulty of training GANs with discrete data by rephrasing the deterministic game in terms of continuous latent variables or simply ignoring the discrete sampling process altogether, and do not directly solve the problem of optimizing the generator from a difference measure estimated from the discriminator.

**Remarks on stabilizing adversarial learning, IPMs, and regularization**    A number of variants of GANs have been introduced recently to address stability issues with GANs. Specifically, generated samples tend to collapse to a set of singular values that resemble the data on neither a per-sample or distribution basis. Several early attempts in modifying the train procedure (Berthelot et al., 2017; Salimans et al., 2016) as well as the identifying of a taxonomy of working architectures (Radford et al., 2015) addressed stability in some limited setting, but it wasn't until Wasserstein GANs (WGAN, Arjovsky et al., 2017) were introduced that there was any significant progress on reliable training of GANs.

WGANs rely on an integral probability metric (IPM, Sriperumbudur et al., 2009) that is the dual to the *Wasserstein distance*. Other GANs based on IPMs, such as Fisher GAN (Mroueh & Sercu, 2017) tout improved stability in training. In contrast to GANs based on $f$-divergences, besides being based on *metrics* that are "weak", IPMs rely on restricting $\mathcal{T}$ to a subset of all possible functions. For instance in WGANs, $\mathcal{T} = \{T \mid \|T\|_L \leq K\}$, is the set of K-Lipschitz functions. Ensuring a statistic network, $T_\phi$, with a large number of parameters is Lipschitz-continuous is *hard*, and these methods rely on some sort of regularization to satisfy the necessary constraints. This includes the original formulation of WGANs, which relied on weight-clipping, and a later work (Gulrajani et al., 2017) which used a gradient penalty over interpolations between real and generated data.

Unfortunately, the above works provide little details on whether $T_\phi$ is actually in the constrained set in practice, as this is probably very hard to evaluate in the high-dimensional setting. Recently, Roth et al. (2017) introduced a gradient norm penalty similar to that in Gulrajani et al. (2017) without interpolations and which is formulated in terms of $f$-divergences. In our work, we've found that this approach greatly improves stability, and we use it in nearly all of our results. That said, it is still unclear empirically how the discriminator objective plays a strong role in stabilizing adversarial learning, but at this time it appears that correctly regularizing the discriminator is sufficient.

## 4 DISCRETE VARIABLES: EXPERIMENTS AND RESULTS

### 4.1 ADVERSARIAL CLASSIFICATION

We first verify the gradient estimator provided by BGAN works quantitatively in the discrete setting by evaluating its ability to train a classifier with the CIFAR-10 dataset (Krizhevsky & Hinton, 2009). The "generator" in this setting is a multinomial distribution, $g_\theta(y \mid x)$ modeled by the softmax output of a neural network. The discriminator, $T_\phi(x, y)$, takes as input an image / label pair so that the variational lower-bound is:

$$\mathcal{V}(\mathbb{P}_{XY}, \mathbb{Q}_{Y|X}\mathbb{P}_X, T_\phi) = \mathbb{E}_{p(x,y)}[T_\phi(x, y)] - \mathbb{E}_{g_\theta(y|x)p(x)}[f^\star(T_\phi(x, y))] \tag{14}$$

For these experiments, we used a simple 4-layer convolutional neural network with an additional 3 fully-connected layers. We trained the importance sampling BGAN on the set of $f$-divergences given in Table 1 as well as the REINFORCE counterpart for 200 epochs and report the accuracy on the test set. In addition, we ran a simple classification baseline trained on cross-entropy as well as a continuous approximation to the problem as used in WGAN-based approaches (Gulrajani et al., 2017). No regularization other than batch normalization (BN, Ioffe & Szegedy, 2015) was used with the generator, while gradient norm penalty (Roth et al., 2017) was used on the statistic networks. For WGAN, we used clipping, and chose the clipping parameter, the number of discriminator updates, and the learning rate separately based on training set performance. The baseline for the REIN-FORCE method was learned using a moving average of the reward.

Table 2: Adversarial classification on CIFAR-10. All methods are BGAN with importance sampling (left) or REINFORCE (right) except for the baseline (cross-entropy) and Wasserstein GAN (WGAN)

| | Measure | Error(%) | |
|---|---|---|---|
| | Baseline | 26.6 | |
| | WGAN (clipping) | 72.3 | |
| | | IS | REINFORCE |
| BGAN | GAN | 26.2 | 27.1 |
| | Jensen-Shannon | 26.0 | 27.7 |
| | KL | 28.1 | 28.0 |
| | Reverse KL | 27.8 | 28.2 |
| | Squared-Hellinger | 27.0 | 28.0 |

Our results are summarized in Table 2. Overall, BGAN performed similarly to the baseline on the test set, with the REINFORCE method performing only slightly worse. For WGAN, despite our best efforts, we could only achieve an error rate of 72.3% on the test set, and this was after a total of 600 epochs to train. Our efforts to train WGAN using gradient penalty failed completely, despite it working with higher-dimension discrete data (see Appendix).

### 4.2 DISCRETE IMAGE AND NATURAL LANGUAGE GENERATION

**Image data: binary MNIST and quantized CelebA**   We tested BGAN using two imaging benchmarks: the common discretized MNIST dataset (Salakhutdinov & Murray, 2008) and a new quantized version of the CelebA dataset (see Liu et al., 2015, for the original CelebA dataset).

For CelebA quantization, we first downsampled the images from $64 \times 64$ to $32 \times 32$. We then generated a 16-color palette using Pillow, a fork of the Python Imaging Project (https://python-pillow.org). This palette was then used to quantize the RGB values of the CelebA samples to a one-hot representation of 16 colors. Our models used deep convolutional GANs (DCGAN, Radford et al., 2015). The generator is fed a vector of 64 i.i.d. random variables drawn from a uniform distribution, $[0, 1]$. The output nonlinearity was sigmoid for MNIST to model the Bernoulli centers for each pixel, while the output was softmax for quantized CelebA.

Our results show that training the importance-weighted BGAN on discrete MNIST data is stable and produces realistic and highly variable generated handwritten digits (Figure 1). Further quantitative experiments comparing BGAN against WGAN with the gradient penalty (WGAN-GP Gulrajani et al., 2017) showed that when training a new discriminator on the samples directly (keeping the

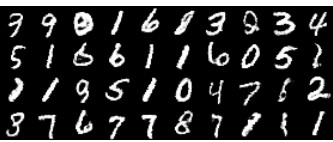

Figure 1: Left: Random samples from the generator trained as a boundary-seeking GAN (BGAN) with discrete MNIST data. Shown are the Bernoulli centers of the generator conditional distribution.

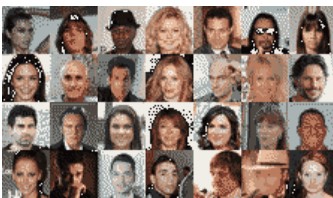 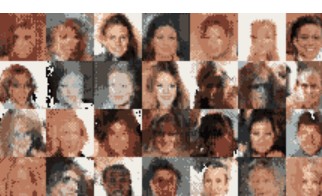

Figure 2: Left: Ground-truth 16-color (4-bit) quantized CelebA images downsampled to $32 \times 32$. Right: Samples produced from the generator trained as a boundary-seeking GAN on the quantized CelebA for 50 epochs.

Table 3: Random samples drawn from a generator trained with the discrete BGAN objective. The model is able to successfully learn many important character-level English language patterns.

| | | |
|---|---|---|
| And it 's miant a quert could he | He weirst placed produces hopesi | What 's word your changerg bette |
| " We pait of condels of money wi | Sance Jory Chorotic , Sen doesin | In Lep Edger 's begins of a find", |
| Lankard Avaloma was Mr. Palin , | What was like one of the July 2 | " I stroke like we all call on a |
| Thene says the sounded Sunday in | The BBC nothing overton and slea | With there was a passes ipposing |
| About dose and warthestrinds fro | College is out in contesting rev | And tear he jumped by even a roy |

generator fixed), the final estimated distance measures were *higher* (i.e., worse) for WGAN-GP than BGAN, *even when comparing using the Wasserstein distance*. The complete experiment and results are provided in the Appendix. For quantized CelebA, the generator trained as a BGAN produced reasonably realistic images which resemble the original dataset well and with good diversity.

**1-billion word** Next, we test BGAN in a natural language setting with the 1-billion word dataset (Chelba et al., 2013), modeling at the character-level and limiting the dataset to sentences of at least 32 and truncating to 32 characters. For character-level language generation, we follow the architecture of recent work (Gulrajani et al., 2017), and use deep convolutional neural networks for both the generator and discriminator.

Training with BGAN yielded stable, reliably good character-level generation (Table 3), though generation is poor compared to recurrent neural network-based methods (Sutskever et al., 2011; Mikolov, 2012). However, we are not aware of any previous work in which a discrete GAN, without any continuous relaxation (Gulrajani et al., 2017), was successfully trained from scratch without pretraining and without an auxiliary supervised loss to generate any sensible text. Despite the low quality of the text relative to supervised recurrent language models, the result demonstrates the stability and capability of the proposed boundary-seeking criterion for training discrete GANs.

## 5 CONTINUOUS VARIABLES: EXPERIMENTS AND RESULTS

Here we present results for training the generator on the boundary-seeking objective function. In these experiments, we use the original GAN variational lower-bound from Goodfellow et al. (2014), only modifying the generator function. All results use gradient norm regularization (Roth et al., 2017) to ensure stability.

### 5.1 GENERATION BENCHMARKS

We test here the ability of continuous BGAN to train on high-dimensional data. In these experiments, we train on the CelebA, LSUN (Yu et al., 2015) datasets, and the 2012 ImageNet dataset with all 1000 labels (Krizhevsky et al., 2012). The discriminator and generator were both modeled as 4-layer Resnets (He et al., 2016) without conditioning on labels or attributes.

Figure 3 shows examples from BGAN trained on these datasets. Overall, the sample quality is very good. Notably, our Imagenet model produces samples that are high quality, despite not being trained

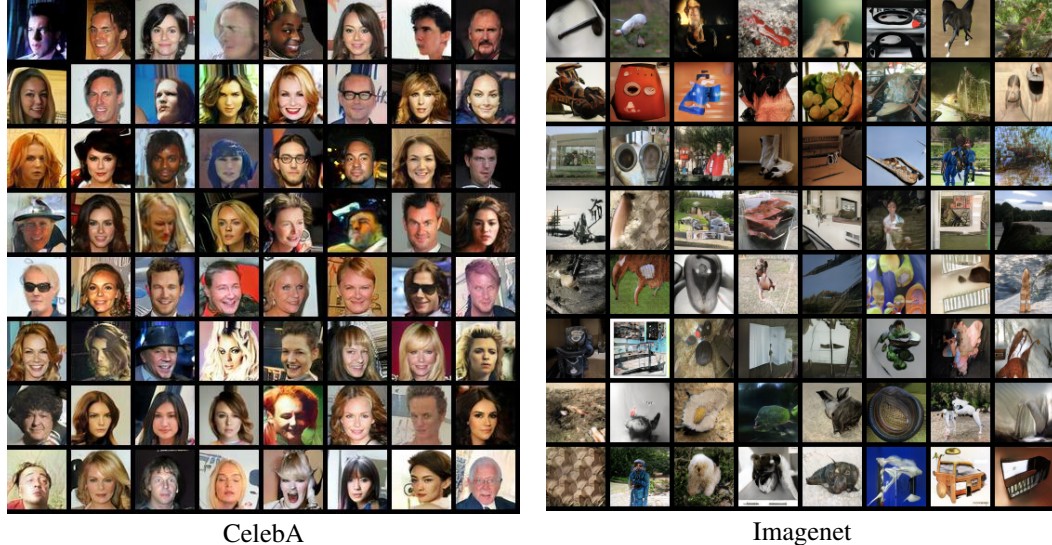

CelebA

Imagenet

LSUN

Figure 3: Highly realistic samples from a generator trained with BGAN on the CelebA and LSUN datasets. These models were trained using a deep ResNet architecture with gradient norm regularization (Roth et al., 2017). The Imagenet model was trained on the full 1000 label dataset without conditioning.

conditioned on the label and on the full dataset. However, the story here may not be that BGAN necessarily generates better images than using the variational lower-bound to train the generator, since we found that images of similar quality on CelebA could be attained without the boundary-seeking loss as long as gradient norm regularization was used, rather we confirm that BGAN works well in the high-dimensional setting.

## 5.2   STABILITY OF CONTINUOUS BGAN

As mentioned above, gradient norm regularization greatly improves stability and allows for training with very large architectures. However, training still relies on a delicate balance between the generator and discriminator: over-training the generator may destabilize learning and lead to worse results. We find that the BGAN objective is resilient to such over-training.

**Stability in training with an overoptimized generator**   To test this, we train on the CIFAR-10 dataset using a simple DCGAN architecture. We use the original GAN objective for the discriminator, but vary the generator loss as the variational lower-bound, the proxy loss (i.e., the generator loss function used in Goodfellow et al., 2014), and the boundary-seeking loss (BGAN). To better study the effect of these losses, we update the generator for 5 steps for every discriminator step.

Our results (Figure 4) show that over-optimizing the generator significantly degrades sample quality. However, in this difficult setting, BGAN learns to generate reasonable samples in fewer epochs than other objective functions, demonstrating improved stability.

**Following the generator gradient**   We further test the different objectives by looking at the effect of gradient descent on the pixels. In this setting, we train a DCGAN (Radford et al., 2015) using the proxy loss. We then optimize the discriminator by training it for another 1000 updates. Next, we perform gradient descent directly on the pixels, the original variational lower-bound, the proxy, and the boundary seeking losses separately.

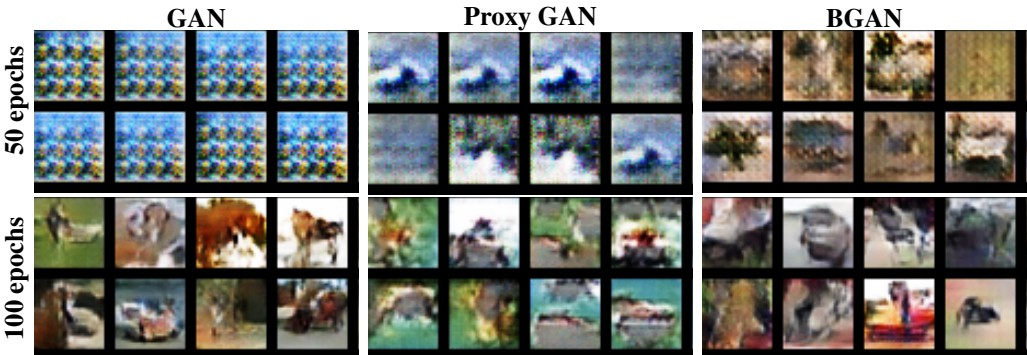

Figure 4: Training a GAN with different generator loss functions and 5 updates for the generator for every update of the discriminator. Over-optimizing the generator can lead to instability and poorer results depending on the generator objective function. Samples for GAN and GAN with the proxy loss are quite poor at 50 discriminator epochs (250 generator epochs), while BGAN is noticeably better. At 100 epochs, these models have improved, though are still considerably behind BGAN.

Our results show that following the BGAN objective at the pixel-level causes the least degradation of image quality. This indicates that, in training, the BGAN objective is the least likely to disrupt adversarial learning.

## 6    CONCLUSION

Reinterpreting the generator objective to match the proposal target distribution reveals a novel learning algorithm for training a generative adversarial network (GANs, Goodfellow et al., 2014). This proposed approach of boundary-seeking provides us with a unified framework under which learning algorithms for both discrete and continuous variables are derived. Empirically, we verified our approach quantitatively and showed the effectiveness of training a GAN with the proposed learning algorithm, which we call a boundary-seeking GAN (BGAN), on both discrete and continuous variables, as well as demonstrated some properties of stability.

## ACKNOWLEDGEMENTS

RDH thanks IVADO, MILA, UdeM, NIH grants R01EB006841 and P20GM103472, and NSF grant 1539067 for support. APJ thanks UWaterloo, Waterloo AI lab and MILA for their support and Michael Noukhovitch, Pascal Poupart for constructive discussions. KC thanks AdeptMind, TenCent, eBay, Google (Faculty Awards 2015, 2016), NVIDIA Corporation (NVAIL) and Facebook for their support. YB thanks CIFAR, NSERC, IBM, Google, Facebook and Microsoft for their support. We would like to thank Simon Sebbagh for his input and help with Theorem 2. Finally, we wish to thank the developers of Theano (Al-Rfou et al., 2016), Lasagne http://lasagne.readthedocs.io, and Fuel (Van Merriënboer et al., 2015) for their valuable code-base.

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

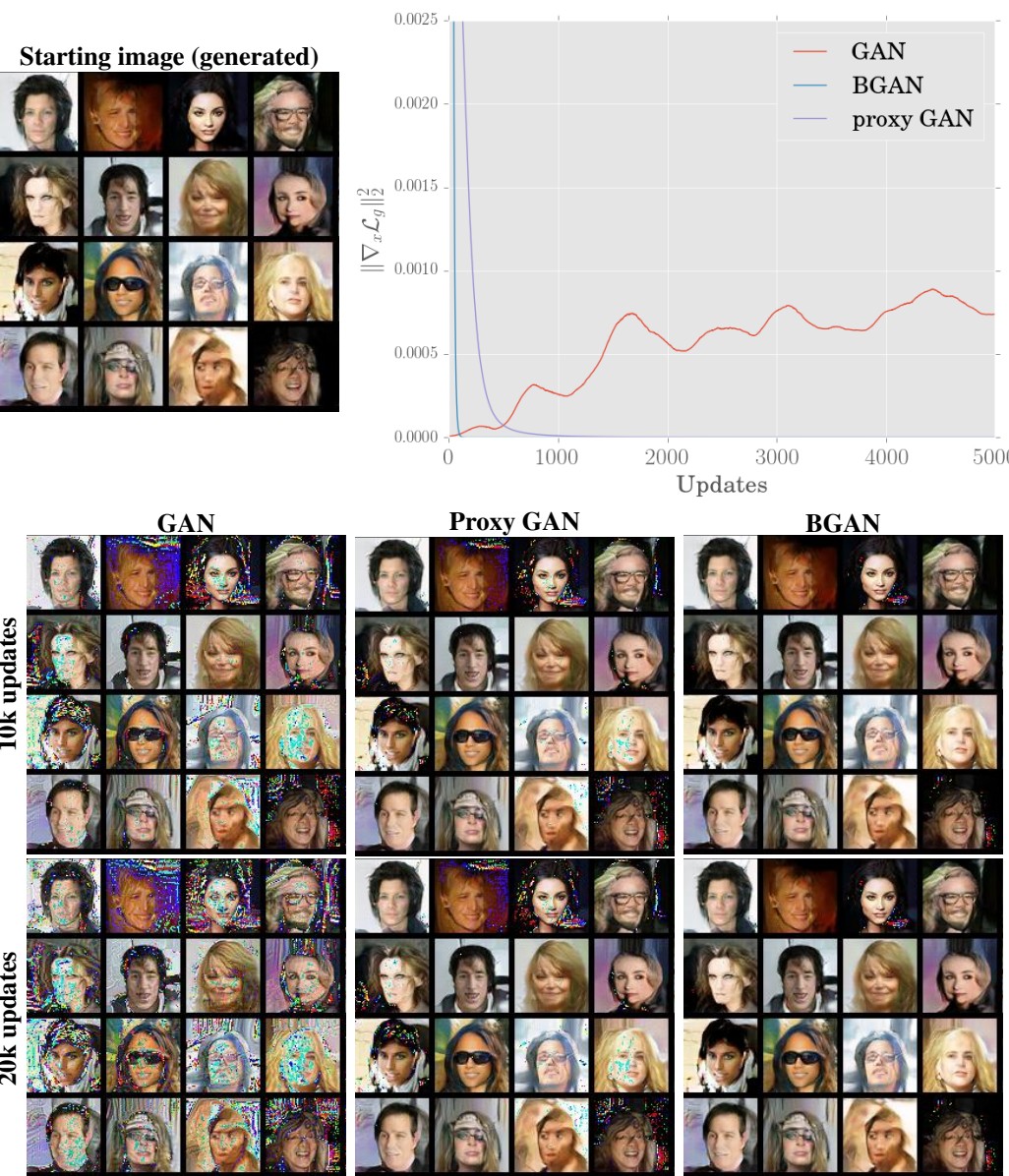

Figure 5: Following the generator objective using gradient descent on the pixels. BGAN and the proxy have sharp initial gradients that decay to zero quickly, while the variational lower-bound objective gradient slowly increases. The variational lower-bound objective leads to very poor images, while the proxy and BGAN objectives are noticeably better. Overall, BGAN performs the best in this task, indicating that its objective will not overly disrupt adversarial learning.

Berthelot, David, Schumm, Tom, and Metz, Luke. Began: Boundary equilibrium generative adversarial networks. *arXiv preprint arXiv:1703.10717*, 2017.

Bornschein, Jörg and Bengio, Yoshua. Reweighted wake-sleep. *arXiv preprint arXiv:1406.2751*, 2014.

Che, Tong, Li, Yanran, Zhang, Ruixiang, Hjelm, R Devon, Li, Weijie, Song, Yangqiu, and Bengio, Yoshua. Maximum-likelihood augmented discrete generative adversarial networks. *arXiv preprint*, 2017.

Chelba, Ciprian, Mikolov, Tomas, Schuster, Mike, Ge, Qi, Brants, Thorsten, Koehn, Phillipp, and Robinson, Tony. One billion word benchmark for measuring progress in statistical language

modeling. *arXiv preprint arXiv:1312.3005*, 2013.

Dai, Bo, Lin, Dahua, Urtasun, Raquel, and Fidler, Sanja. Towards diverse and natural image descriptions via a conditional gan. *arXiv preprint arXiv:1703.06029*, 2017.

Goodfellow, Ian, Pouget-Abadie, Jean, Mirza, Mehdi, Xu, Bing, Warde-Farley, David, Ozair, Sherjil, Courville, Aaron, and Bengio, Yoshua. Generative adversarial nets. In *Advances in Neural Information Processing Systems*, pp. 2672–2680, 2014.

Gu, Shixiang, Levine, Sergey, Sutskever, Ilya, and Mnih, Andriy. Muprop: Unbiased backpropagation for stochastic neural networks. *arXiv preprint arXiv:1511.05176*, 2015.

Gulrajani, Ishaan, Ahmed, Faruk, Arjovsky, Martin, Dumoulin, Vincent, and Courville, Aaron. Improved training of wasserstein gans. *arXiv preprint arXiv:1704.00028*, 2017.

Gumbel, Emil Julius and Lieblein, Julius. Statistical theory of extreme values and some practical applications: a series of lectures. *US Govt. Print. Office*, 1954.

He, Kaiming, Zhang, Xiangyu, Ren, Shaoqing, and Sun, Jian. Deep residual learning for image recognition. In *Proceedings of the IEEE conference on computer vision and pattern recognition*, pp. 770–778, 2016.

Huang, Gabriel, Berard, Hugo, Touati, Ahmed, Gidel, Gauthier, Vincent, Pascal, and Lacoste-Julien, Simon. Parametric adversarial divergences are good task losses for generative modeling. *arXiv preprint arXiv:1708.02511*, 2018.

Ioffe, Sergey and Szegedy, Christian. Batch normalization: Accelerating deep network training by reducing internal covariate shift. *arXiv preprint arXiv:1502.03167*, 2015.

Jang, Eric, Gu, Shixiang, and Poole, Ben. Categorical reparameterization with gumbel-softmax. *arXiv preprint arXiv:1611.01144*, 2016.

Jie, Tang and Abbeel, Pieter. On a connection between importance sampling and the likelihood ratio policy gradient. In *Advances in Neural Information Processing Systems*, pp. 1000–1008, 2010.

Kim, Yoon, Zhang, Kelly, Rush, Alexander M, LeCun, Yann, et al. Adversarially regularized autoencoders for generating discrete structures. *arXiv preprint arXiv:1706.04223*, 2017.

Krizhevsky, Alex and Hinton, Geoffrey. Learning multiple layers of features from tiny images. *Citeseer*, 2009.

Krizhevsky, Alex, Sutskever, Ilya, and Hinton, Geoffrey E. Imagenet classification with deep convolutional neural networks. In *Advances in neural information processing systems*, pp. 1097–1105, 2012.

Lamb, Alex M, GOYAL, Anirudh Goyal ALIAS PARTH, Zhang, Ying, Zhang, Saizheng, Courville, Aaron C, and Bengio, Yoshua. Professor forcing: A new algorithm for training recurrent networks. In *Advances In Neural Information Processing Systems*, pp. 4601–4609, 2016.

Li, Jiwei, Monroe, Will, Shi, Tianlin, Ritter, Alan, and Jurafsky, Dan. Adversarial learning for neural dialogue generation. *arXiv preprint arXiv:1701.06547*, 2017.

Liu, Ziwei, Luo, Ping, Wang, Xiaogang, and Tang, Xiaoou. Deep learning face attributes in the wild. In *Proceedings of the IEEE International Conference on Computer Vision*, pp. 3730–3738, 2015.

Maddison, Chris J, Mnih, Andriy, and Teh, Yee Whye. The concrete distribution: A continuous relaxation of discrete random variables. *arXiv preprint arXiv:1611.00712*, 2016.

Mao, Xudong, Li, Qing, Xie, Haoran, Lau, Raymond YK, Wang, Zhen, and Smolley, Stephen Paul. Least squares generative adversarial networks. *arXiv preprint ArXiv:1611.04076*, 2016.

Mikolov, Tomáš. *Statistical Language Models Based on Neural Networks*. PhD thesis, Ph. D. thesis, Brno University of Technology, 2012.

Mnih, Andriy and Gregor, Karol. Neural variational inference and learning in belief networks. In *Proceedings of the 31st International Conference on Machine Learning (ICML-14)*, pp. 1791–1799, 2014.

Mohamed, Shakir and Lakshminarayanan, Balaji. Learning in implicit generative models. *arXiv preprint arXiv:1610.03483*, 2016.

Mroueh, Youssef and Sercu, Tom. Fisher gan. *arXiv preprint arXiv:1705.09675*, 2017.

Mroueh, Youssef, Sercu, Tom, and Goel, Vaibhava. Mcgan: Mean and covariance feature matching gan. *arXiv preprint arXiv:1702.08398*, 2017.

Nguyen, XuanLong, Wainwright, Martin J, and Jordan, Michael I. Estimating divergence functionals and the likelihood ratio by convex risk minimization. *IEEE Transactions on Information Theory*, 56(11):5847–5861, 2010.

Nowozin, Sebastian, Cseke, Botond, and Tomioka, Ryota. f-gan: Training generative neural samplers using variational divergence minimization. In *Advances in Neural Information Processing Systems*, pp. 271–279, 2016.

Radford, Alec, Metz, Luke, and Chintala, Soumith. Unsupervised representation learning with deep convolutional generative adversarial networks. *arXiv preprint arXiv:1511.06434*, 2015.

Roth, Kevin, Lucchi, Aurelien, Nowozin, Sebastian, and Hofmann, Thomas. Stabilizing training of generative adversarial networks through regularization. *arXiv preprint arXiv:1705.09367*, 2017.

Rubinstein, Reuven Y and Kroese, Dirk P. *Simulation and the Monte Carlo method*, volume 10. John Wiley & Sons, 2016.

Salakhutdinov, Ruslan and Murray, Iain. On the quantitative analysis of deep belief networks. In *Proceedings of the 25th international conference on Machine learning*, pp. 872–879. ACM, 2008.

Salimans, Tim, Goodfellow, Ian, Zaremba, Wojciech, Cheung, Vicki, Radford, Alec, and Chen, Xi. Improved techniques for training gans. In *Advances in Neural Information Processing Systems*, pp. 2234–2242, 2016.

Sriperumbudur, Bharath K, Fukumizu, Kenji, Gretton, Arthur, Schölkopf, Bernhard, and Lanckriet, Gert RG. On integral probability metrics,\phi-divergences and binary classification. *arXiv preprint arXiv:0901.2698*, 2009.

Sutherland, Dougal J, Tung, Hsiao-Yu, Strathmann, Heiko, De, Soumyajit, Ramdas, Aaditya, Smola, Alex, and Gretton, Arthur. Generative models and model criticism via optimized maximum mean discrepancy. *arXiv preprint arXiv:1611.04488*, 2016.

Sutskever, Ilya, Martens, James, and Hinton, Geoffrey E. Generating text with recurrent neural networks. In *Proceedings of the 28th International Conference on Machine Learning (ICML-11)*, pp. 1017–1024, 2011.

Tran, Dustin, Ranganath, Rajesh, and Blei, David M. Deep and hierarchical implicit models. *arXiv preprint arXiv:1702.08896*, 2017.

Tucker, George, Mnih, Andriy, Maddison, Chris J, and Sohl-Dickstein, Jascha. Rebar: Low-variance, unbiased gradient estimates for discrete latent variable models. *arXiv preprint arXiv:1703.07370*, 2017.

Van Merriënboer, Bart, Bahdanau, Dzmitry, Dumoulin, Vincent, Serdyuk, Dmitriy, Warde-Farley, David, Chorowski, Jan, and Bengio, Yoshua. Blocks and fuel: Frameworks for deep learning. *arXiv preprint arXiv:1506.00619*, 2015.

Yu, Fisher, Zhang, Yinda, Song, Shuran, Seff, Ari, and Xiao, Jianxiong. Lsun: Construction of a large-scale image dataset using deep learning with humans in the loop. *arXiv preprint arXiv:1506.03365*, 2015.

Yu, Lantao, Zhang, Weinan, Wang, Jun, and Yu, Yong. Seqgan: sequence generative adversarial nets with policy gradient. *arXiv preprint arXiv:1609.05473*, 2016.

## 7 APPENDIX

### 7.1 COMPARISON OF DISCRETE METHODS

In these experiments, we produce some quantitative measures for BGAN against WGAN with the gradient penalty (WGAN-GP, Gulrajani et al., 2017) on the discrete MNIST dataset. In order to use back-propagation to train the generator, WGAN-GP uses the *softmax probabilities* directly, bypassing the sampling process at pixel-level and problems associated with estimating gradients through discrete processes. Despite this, WGAN-GP is been able to produce samples that visually resemble the target dataset.

Here, we train 3 models on the discrete MNIST dataset using identical architectures with the BGAN with the JS and reverse KL $f$-divergences and WGAN-GP objectives. Each model was trained for 300 generator epochs, with the discriminator being updated 5 times per generator update for WGAN-GP and 1 time per generator update for the BGAN models (in other words, the generators were trained for the same number of updates). This model selection procedure was chosen as the difference measure (i.e., JSD, reverse KL divergence, and Wasserstein distance) as estimated during training converged for each model. WGAN-GP was trained with a gradient penalty hyper-parameter of 5.0, which did not differ from the suggested 10.0 in our experiments with discrete MNIST. The BGAN models were trained with the gradient norm penalty of 5.0 (Roth et al., 2017).

Next, for each model, we trained 3 new discriminators with double capacity (twice as many hidden units on each layer) to maximize the the JS and reverse KL divergences and Wasserstein distance, keeping the generators fixed. These discriminators were trained for 200 epochs (chosen from convergence) with the same gradient-based regularizations as above. For all of these models, the discriminators were trained using the *samples*, as they would be used in practical applications. For comparison, we also trained an additional discriminator, evaluating the WGAN-GP model above on the Wasserstein distance using the softmax probabilities.

Table 4: Estimated Jensen-Shannon and KL-divergences and Wasserstein distance by a discriminator trained to maximize the respective lowerbound (lower is better). Numbers are estimates averaged ovwe 12 batches of 5000 samples with standard devations provided in parentheses. All discriminators were trained using samples drawn from the softmax probabilities, with exception to an additional discriminator used to evaluate WGAN-GP where the softmax probabilities were used directly. In general, BGAN out-performs WGAN-GP even when comparing the Wasserstein distances.

| Train Measure | Eval Measure (lower is better) | | |
|---|---|---|---|
| | JS | reverse KL | Wasserstein |
| BGAN - JS | 0.37 ($\pm$0.02) | 0.16 ($\pm$0.01) | 0.40 ($\pm$0.03) |
| BGAN - reverse KL | 0.44 ($\pm$0.02) | 0.44 ($\pm$0.03) | 0.45 ($\pm$0.04) |
| WGAN-GP (samples) | 0.45 ($\pm$0.03) | 1.32 ($\pm$0.06) | 0.87 ($\pm$0.18) |
| WGAN-GP (softmax) | - | - | 0.54 ($\pm$0.12) |

Final evaluation was done by estimating difference measures using 60000 MNIST training examples againt 60000 samples from each generator, averaged over 12 batches of 5000. We used the training set as this is the distribution over which the discriminators were trained. Test set estimates in general were close and did not diverge from training set distances, indicating the discriminators were not overfitting, but training set estimates were slightly higher on average.

Our results show that the estimates from the sampling distribution from BGAN is consistently lower than that from WGAN-GP, *even when evaluating using the Wasserstein distance*. However, when training the discriminator on the softmax probabilities, WGAN-GP has a much lower Wasserstein distance. Despite quantitative differences, samples from these different models were indistinguishable as far as quality by visual inspection. This indicates that, though playing the adversarial game using the softmax outputs can generate realistic-looking samples, this procedure ultimately hurts the generator's ability to model a truly discrete distribution.

## 7.2 THEORETICAL AND EMPIRICAL VALIDATION OF THE VARIANCE REDUCTION METHOD

Here we validate the policy gradient provided in Equation 10 theoretically and empirically.

**Theorem 2.** *Let the expectation of the conditional KL-divergence be defined as in Equation 9. Then* $\mathbb{E}_{h(z)}[\mathcal{D}_{KL}\left(\tilde{p}(x \mid z)\|g_\theta(x \mid z)\right)] = 0 \implies \mathcal{D}_{KL}(\tilde{p}(x)\|q_\theta) = 0.$

*Proof.* As the conditional KL-divergence is has an absolute minimum at zero, the expectation can only be zero when the all of the conditional KL-divergences are zero. In other words:

$$\mathbb{E}_{h(z)}[\mathcal{D}_{KL}\left(\tilde{p}(x \mid z)\|g_\theta(x \mid z)\right)] = 0 \implies \tilde{p}(x \mid z) = g_\theta(x \mid z). \tag{15}$$

As per the definition of $\tilde{p}(x \mid z)$, this implies that $\alpha(z) = w(x) = C$ is a constant. If $w(x)$ is a constant, then the partition function $\beta = C\mathbb{E}_{\mathbb{Q}_\theta}[1] = C$ is a constant. Finally, when $\frac{w(x)}{\beta} = 1$, $\tilde{p}(x) = q_\theta \implies \mathcal{D}_{KL}(\tilde{p}(x)\|q_\theta) = 0.$ $\qquad\square$

In order to empirically evaluate the effect of using an Monte-Carlo estimate of $\beta$ from Equation 8 versus the variance-reducing method in Equation 10, we trained several models using various sample sizes from the prior, $h(z)$, and the conditional, $g_\theta(x \mid z)$.

We compare both methods with 64 samples from the prior and 5, 10, and 100 samples from the conditional. In addition, we compare to a model that estimates $\beta$ using 640 samples from the prior and a single sample from the conditional. These models were all run on discrete MNIST for 50 epochs with the same architecture as those from Section 4.2 with a gradient penalty of 1.0, which was the minimum needed to ensure stability in nearly all the models.

Our results (Figure 6) show a clear improvement using the variance-reducing method from Equation 10 over estimating $\beta$. Wall-clock times were nearly identical for methods using the same number of total samples (blue, green, and red dashed and solid line pairs). Both methods improve as the number of conditional samples is increased.

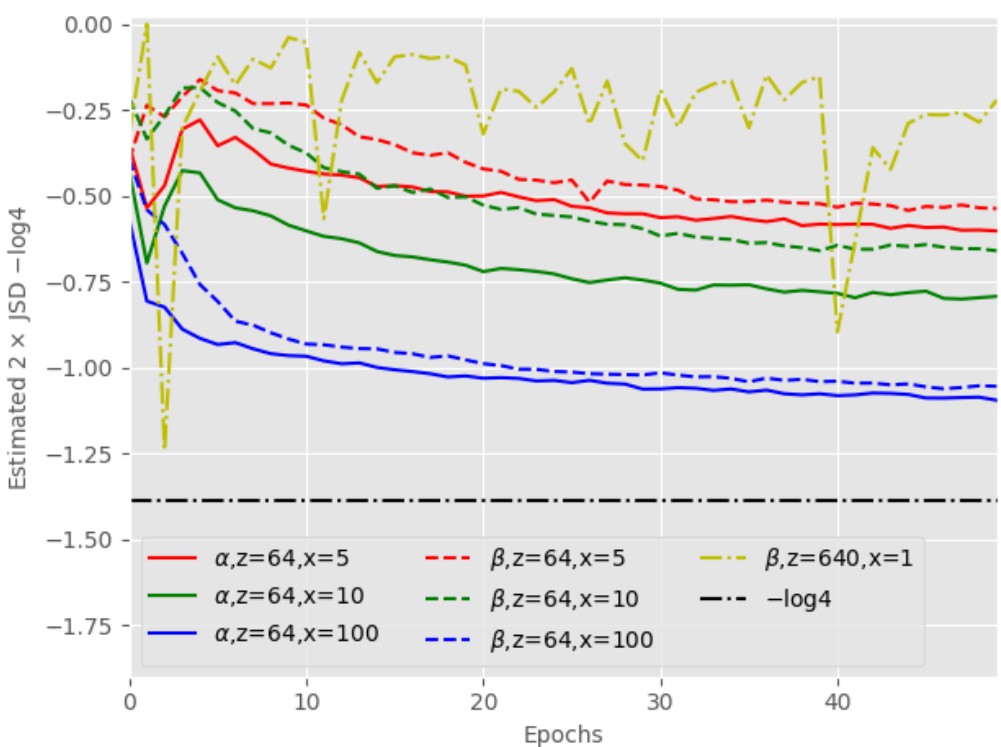

Figure 6: Comparison of the variance-reducing method from Equation 10 and estimating $\beta$ using Monte-Carlo in Equation 8. $\alpha$ indicates the variance-reducing method, and $\beta$ is estimating $\beta$ using Monte-Carlo. $z =$ indicates the number of samples from the prior, $h(z)$, and $x =$ indicates the number of samples from the conditional, $g_\theta(x \mid z)$ used in estimation. Plotted are the estimated GAN distances $(2 * \mathrm{JSD} - \log 4)$ from the discriminator. The minimum GAN distance, $-\log 4$, is included for reference. Using the variance-reducing method gives a generator with consistently lower estimated distances than estimating $\beta$ directly.

