# OpenReview forum: "Boundary Seeking GANs"
_ICLR.cc/2018/Conference — Accept (Poster)_

### Official Review · AnonReviewer3 · 2017-11-27
**Contains some interesting ideas**

**Rating:** 7
**Confidence:** 3

**Review:**

Thank you for the feedback, and I have read the revision.

I would say the revised version has more convincing experimental results (although I'm not sure about the NLP part). The authors have also addressed my concerns on variance reduction, although it's still mysterious to me that the density ratio estimation method seems to work very well even at the begining stage.

Also developing GAN approaches for discrete variables is an important and unsolved problem.

Considering all of the above, I would like to raise the rating to 7, but lower my confidence to 3 (as I'm not an expert for NLP which is the main task for discrete generative models).

==== original review ====

Thank you for an interesting read.

My understanding of the paper is that:

1. the paper proposes a density-ratio estimator via the f-gan approach;
2. the paper proposes a training criterion that matches the generator's distribution to a self-normalised importance sampling (SIS) estimation of the data distribution;
3. in order to reduce the variance of the REINFORCE gradient, the paper seeks out to do matching between conditionals instead.

There are a few things that I expect to see explanations, which are not included in the current version:

1. Can you justify your variance reduction technique either empirically or experimentally? Because your method requires sampling multiple x for a single given z, then in the same wall-clock time I should be able to obtain more samples for the vanilla version eq (8). How do they compare?

2. Why your density ratio estimation methods work in high dimensions, even when at the beginning p and q are so different?

3. It's better to include some quantitative metrics for the image and NLP experiments rather than just showing the readers images and sentences!

4. Over-optimising generators is like solving a max-min problem instead. You showed your method is more robust in this case, can you explain it from the objective you use, e.g. the convex/concavity of your approach in general?

Typo: eq (3) should be min max I believe?

BTW I'm not an expert of NLP so I won't say anything about the quality of the NLP experiment.

---

> ### Author Response · Authors · 2017-12-08
> **Thank you for your comments and suggestions**
>
> You’re welcome: we enjoyed writing it.
> 1) Yes! We should have shown this, at least in the appendix. We can definitely justify the variance reduction technique empirically. Would you like to see an experiment where we compare the estimated likelihood ratio, using equations 8 vs 10 for training and a simple dataset like MNIST, in the main text or the appendix? We can provide results with varying sample sizes and compare against wall-clock time.
>
> 2) This is a very interesting and important question; as we are effectively using importance sampling, if p and q are very different, the unnormalized importance weights will be effectively 0. So why does this work in our case? The hypothesis is that because the pixel values themselves overlap (as they are discrete), that this is enough to create some variation in the importance weights, which is magnified by the normalization. At first, this could encourage the generator to produce pixels that the discriminator likes to see, effectively bringing the two distributions closer together. This should also apply in the high-dimensional setting as even a small overlap will be magnified by the normalization process. We are currently formulating an experiment we believe will help educate the reader a little better about what is going on here, and will include it in the revision.
>
> 3) We agree: and we’re quite unhappy that there currently aren’t any reasonable metrics for evaluating GANs. What sorts of metrics would you like to see? Likelihood estimates are not considered a good metric for performance of GANs (as they aren’t optimized on the likelihood directly), and we’ve found that methods for estimating likelihoods for discrete GANs using AIS (Wu et al 2016) completely failed for us (using their code). We can provide inception scores for MNIST across different f-divergences (and include WGAN + WGAN-GP for some reference). Alternatively, we can use  an estimate of beta, which converges to the likelihood ratio as the discriminator improves, and provide a comparison across different f-divergences (at least for MNIST).
>
> 4) Yes! This is a very nice observation: using the square distance from the decision boundary transforms a max problem over a convex function to one over a concave function. If you don’t mind, we would like to add this insight to the section addressing stability of the generator, as it is a very nice way of thinking of what is going on.
>
> And yes: it should be min max; thank you

---

> > ### Comment · AnonReviewer3 · 2017-12-17
> > **replying your feedback**
> >
> > Thank you for your feedback.
> >
> > Just wanted to say that for GAN papers I would like to see more insights. If you can say more about your method, e.g. the variance reduction technique (how it relates to e.g. Rao-Blackwellization) and the generator over-training, then that would be nice, and I will consider it.
> >
> > Also, quantitative results will be important. I agree there's no rigorous assessment metric here (especially that I am not familiar with NLP), but for the natural image experiments, maybe inception score and other recently proposed metrics like FID can be helpful.
> >
> > Let me know if you have your revision ready.

---

> ### Author Response · Authors · 2017-12-31
> **Revision**
>
> Following your concerns, we have added a section to the Appendix that introduces an experiment comparing the estimated GAN-distance during training across models trained by the variance-reducing method (eq 10) and models trained by estimating beta (eqs 7, 8) using Monte Carlo. From these experiments, we are able to make the following conclusions: a) more samples from the conditional in training achieves lower GAN-divergence (2 JSD - 2log4) and b) eq 10 consistently achieves lower GAN-divergence than using MC estimate of beta from eq 8. Next, we added your insight regarding the convex/concavity of using the square error loss in the continuous case (see next to last paragraph, page 6, starting with “This objective can be seen”).
>
> Next, rather than using Inception score, which has not been used for quantitative experiments with discrete data, we trained new discriminators with higher capacities to estimate the Wasserstein distance or f-divergences between the MNIST training set and the discrete generated samples (keeping the generators fixed). Our results show that BGAN outperforms WGAN-GP consistently across *all metrics*, including the Wasserstein distance. Though we cannot say with absolutely certainty, it is very likely that this is because, while WGAN-GP is able to generate samples that visually resemble the target dataset, using the softmax outputs hurts the generator’s ability to model a truly discrete distribution. Please refer to the Appendix, section 7.1 for details.
>
> We attempted some experiments to illuminate why BGAN works using importance sampling, especially at the beginning of training when we know the distribution overlap will be small. However, at this time, we do not have any results that would paint a particularly clear picture to the reader. For instance, when looking at the effective sample size over the first epoch, we found this quantity fluctuates rapidly at the beginning of training, until converging to a reasonable value (about 90%). We expect that the effective sample size might be highly variable in the beginning of training as the unnormalized weights as very low, but not uniform. However, this is still speculation, and we believe that more time and care is needed to make any conclusions in the text.

---

### Official Review · AnonReviewer1 · 2017-11-27
**review for boundary seeking GAN**

**Rating:** 7
**Confidence:** 4

**Review:**

Thanks for the feedback and for clarifying the 1) algorithm and the assumptions in the multivariate case 2) comparison to RL based methods 3) connection to estimating importance sampling weights using GAN discriminator.

I think the paper contribution is now more clear and strengthened with additional convincing experiments and I am increasing my score to 7.

The paper would still benefit from doing the experiment with importance weights by pixel , rather then a global one as done in the paper now. I encourage the authors to still do the experiment, see if there is any benefit.



==== Original Review =====
Summary of the paper:

The paper presents a method based on importance sampling and reinforcement learning to learn discrete generators in the GAN framework. The GAN uses an  f-divergence cost function for  training the discriminator. The generator is trained to minimize the KL distance between the  discrete generator q_{\theta}(x|z), and the importance weight discrete real distribution estimator w(x|z)q(\theta|z). where w(x|z) is estimated in turn using the discriminator.
The methodology is also extended to the continuous case. Experiments are conducted on quantized image generation, and text generation.

Quality:

the paper is overall well written and supported with reasonable experiments.

Clarity:

The paper has a lot of typos that make sometimes the paper harder to follow:
- page (2) Eq 3 max , min should be min, max if we want to keep working with f-divergence
- Definition 2.1 \mathbb{Q}_{\theta} --> \mathbb{Q}
- page 5 the definition of \tilde{w}(x^{(m})) in the normalization it is missing \tilde{w}
- Equation (10) \nabla_{\theta}\log(x|z) --> \nabla_{\theta}\log(x^{(m)}|z)
- In algorithm 1, again missing indices in the update of theta  --> \nabla_{\theta}\log(x^{(m|n)}|z^{n})

Originality:

The main ingredients of the paper are well known and already used in the literature (Reinforce for discrete GAN with Disc as a reward for e.g GAN for image captioning Dai et al). The perspective from importance sampling coming from f-divergence for discrete GAN has some novelty although the foundations of this work relate also to previous work:
- Estimating ratios using the discriminator is well known for e.g learning implicit models , Mohamed et al
- The relation of  importance sampling to  reinforce is also well known" On a Connection between Importance Sampling and the Likelihood Ratio Policy Gradient," Tang and Abbeel.

General Review:

- when the generator is producing only *one* discrete distribution the theory is presented in Section 2.3. When we move to experiments, for image generation for example, we need to have a generator that produces a distribution by pixel. It would be important for 1) understanding the work 2) the reproducibility of the work to parallel algorithm 1 and have it *in the paper*, for this 'multi discrete distribution ' generation case.  If we have N pixels    \log(p(x_1,...x_N|z))= \Pi_i g_{\theta}(x_i|z) (this should be mentioned in the paper if it is the case ), it would be instructive to comment on the assumptions on independence/conditional dependence of this model, also to state clearly how the generator is updated in this case and what are importance sampling weights.

- Would it make sense in this N pixel discrete case generation to have also the discriminator produce N probabilities of real and fake as in PixelGAN in Isola et al? then see in this case what are the importance sampling weights this would parallel the instantaneous reward in RL?

---

> ### Author Response · Authors · 2017-12-08
> **Thank you for your comments**
>
> You’re absolutely right: the observations made in our paper are not completely unknown to the community. Another paper we failed to mention in our work is Tran et al “Hierarchical Implicit Models and Likelihood-Free Variational Inference”, which also uses the estimate of the likelihood ratio in learning. Our primary contribution is connecting all of these ideas and present a principled method for training GANs on discrete data. The methods that are doing reinforce are actually doing the right thing: using the discriminator output as the score corresponds to using the sigmoid of the log ratio estimate as the reward signal. However, to our knowledge, none of those works made the actual connection to the likelihood ratio estimate in their motivation / formulation. Most of them focus instead on the difficulties associated with language modeling (roll-out policies, actor-critic, MC search, etc), and our family of policy gradients are compatible with many of these.
>
> We think it would be worth adding a paragraph to the related works sections summarizing other works that have made similar observations, like the ones you mention. Do you think this would be a positive addition to the paper for the revision?
>
> As far as the movement from one to the multi discrete distributions: yes, you are correct. We are indeed assuming conditional independence in the generator output variables (conditioned on z). We should have mentioned that in the text and the algorithm and will make the changes you suggest in the revision.
>
> The generator is trained applying the global importance weight derived from the discriminator (we will clarify this in the text) uniformly across all of the pixels. This raises several questions about doing correct credit assignment across the pixels. Using PixelGAN or PatchGAN would be one way to extend our method to do so. Each of the patch or pixel discriminators in this case would be estimating the likelihood ratios for the respective pixel distributions, which can be used to provide variable importance weights across the whole image. One could even construct a hierarchy of these things, so that you had importance weights on both the local and the global scale. We considered these as extensions of our method, but decided that they were out of scope of this particular work.
>
> Yes, it should be min max (and thank you for pointing out some other typos, we really appreciate it).

---

> ### Author Response · Authors · 2017-12-31
> **Revision**
>
> We have added citations on other works that have explored the connection between the discriminator output and likelihood ratios, adding a paragraph in the Related Works section dedicated to this (see “On estimating likelihood ratios from the discriminator"). We have also strengthened references and comparisons to other methods for training GANs with REINFORCE (see “GAN for discrete variables"). Finally, we have clarified that when the data distribution is multivariate (such as with pixels), we assume the observed variables are independent conditioned on Z (see the next to last paragraph of page 5, starting with “Algorithm 1” as well as Algorithm 1).

---

> ### Author Response · Authors · 2018-01-12
> **Revision reminder**
>
> We would like to kindly remind the reviewer of our revision. The complete revision list is provided in the main thread (titled, "Revision available").

---

> > ### Comment · AnonReviewer1 · 2018-01-12
> > **Thank you**
> >
> > I updated my review and increased my score to 7.

---

### Official Review · AnonReviewer2 · 2017-11-27
**Why should I use your method?**

**Rating:** 4
**Confidence:** 3

**Review:**

The paper introduces a new method for training GANs with discrete data. To this end, the output of the discriminator is interpreted as importance weight and REINFORCE-like updates are used to train the generator.

Despite making interesting connections between different ideas in GAN training, I found the paper to be disorganized and hard to read. My main concern is the fact that the paper does not make any comparison with other methods for handling of discrete data in GANs. In particular, (Gulrajani et al.’17) show that it is possible to train Wasserstein GANs without sampling one-hot vectors for discrete variables during the training. Is there a reason to use REINFORCE-like updates when such a direct approach works?

Minor:
complex conjugate => convex conjugate

---

> ### Author Response · Authors · 2017-12-08
> **Because WGAN-GP with softmax relaxation is not a serious solution to a difficult and important problem**
>
> First, we actually *did* compare to another method, namely WGAN-GP with the softmax relaxation in the adversarial classifier experiment. We didn’t include WGAP-GP in the table because, as stated in the text, “Our efforts to train WGAN using gradient penalty failed completely”. So effectively, the error rate was 90% (chance), despite trying *very hard* to get it to work and using a wide variety of regularization hyper-parameters, learning rates, and training ratios. We even consulted the original authors, but they never confirmed whether it worked for them. So if WGAN-GP with continuous relaxation doesn’t work in this simple setting, why would you trust it with something more complicated?
>
> Other than WGAN-GP and ours, no other method has shown to “work” (e.g., Gumbel softmax). Some other works, e.g., Li et al, use the discriminator output to do REINFORCE; which is just a scaled version of our REINFORCE version of BGAN, namely sigma log w (we will add this insight to the revision). So the REINFORCE-based estimators we compare to in the classification experiment are doing *essentially the same thing*. But none of the other works that do this establish why this is a good idea: the motivation is pure RL.
>
> What sort of comparison would you like to see beyond this? If it’s simple enough and would help educate your decision, we would be more than happy to add it to the revision in a timely manner. Perhaps we can provide inception scores for MNIST across different f-divergences and include WGAN + WGAN-GP for some reference?
>
> In some sense though, the answer to your question depends on your personal values and background as a researcher. There are a great number of researchers who find WGAN-GP unsatisfying / problematic (including us) because it’s training a model to match the softmax probabilities with one-hot vectors (apples to oranges). Some of the problems with this are covered in some detail in the now (in)famous blog post on followup work for NLP (Adversarial generation of natural language):
> https://medium.com/@yoav.goldberg/an-adversarial-review-of-adversarial-generation-of-natural-language-409ac3378bd7
> To summarize this, WGAN-GP using the softmax outputs directly is a sort of an unprincipled and “easy” way to get around the back-prop through discrete sampling issue. Some people have shown it “works” (generates some sensible text) with language, but the results are so far so bad compared to MLE that no NLP person takes them seriously. The approach brings up many questions about why it *should* even learn the true distribution, but the authors never even bother to attempt to answer these question reasonably.
>
> Besides, WGAN has biased gradients (see Cramer GAN), so there’s that.
>
> In summary to answer your question: if you want a method with little theoretical backing but is “easy” in the sense that you can plug it into your standard deep learning library and just train using back-prop, then BGAN is not for you. However, if you want a *principled* method that provably converges but requires a little more effort to code, then BGAN provides a strong set of tools for solving your problem in a sensible way.
>
> But we implore you: please reconsider the significance of our work w.r.t. the larger research effort to do adversarial learning with discrete variables. We didn’t write this paper to show that we had “solved” NLP or to even cover the fundamental flaws associated with WGAN-GP with continuous relaxation. We wanted to offer a solution to a fundamentally hard and unsolved problem, and we were excited about the connections to other works we found along the way and felt it was worth sharing.
>
> As far as the organization: perhaps you felt the paper was a little TL;DR (nearly 11 pages)? If so, we apologize about the length: we could have done the usual tricks to bring the paper length down: half the size of the figures, remove the headers such as “definition” removed the theorem/proof structure. But the choices we made we felt would improve the clarity of the piece (and indeed one of the reviewers applauds the quality, and both of the other reviewers clearly understand the paper well and had no issues with the quality). But it is easy enough to misjudge what an average reader might find is clear. What would make the paper easier to read, in your opinion? Should we move some of the definitions and proofs to the appendix?

---

> ### Author Response · Authors · 2017-12-31
> **BGAN vs WGAN-GP quantitative results on discrete MNIST**
>
> In order to better and more definitively answer your question regarding why to use BGAN as well as address your concerns of a relative lack of comparison to other methods, we have included an additional quantitative experiment comparing BGAN to WGAN-GP with discrete MNIST, a summary of which is in the main text with the full details provided in the Appendix, section 7.1. Rather than using Inception score, which has not been used for quantitative experiments with discrete data, we trained new discriminators with higher capacities to estimate the Wasserstein distance or f-divergences between the MNIST training set and the discrete generated samples (keeping the generators fixed). Our results show that BGAN outperforms WGAN-GP consistently across *all metrics*, including the Wasserstein distance. Though we cannot say with absolutely certainty, it is very likely that this is because, while WGAN-GP is able to generate samples that visually resemble the target dataset, using the softmax outputs hurts the generator’s ability to model a truly discrete distribution. Please refer to the Appendix, section 7.1 for details.

---

> ### Author Response · Authors · 2018-01-12
> **Revision reminder**
>
> We would like to kindly remind the reviewer of our revision. The complete revision list is provided in the main thread (titled, "Revision available").

---

### Author Response · Authors · 2017-12-08
**Imagenet results**

Just a global comment to modify the individual comments below:

We recently got very good results on training on full ImageNet 2012, without conditioning on the label, (e.g., not using conditional GANs or auxiliary classifier GANs), using our method. To our knowledge, these are the best ImageNet results compared to those available in the literature. However, while the images are very high-quality, diverse, and show no evidence of mode collapse, most resemble deformations of real things (though the background is usually very good). Would you consider this a positive addition to the paper?

---

### Author Response · Authors · 2017-12-31
**Revision available**

Dear reviewers, we have submitted a revision to our paper, "Boundary Seeking GANs" (we mistakenly submitted it twice, so please ignore one of the last two revisions when you build the diff pdf). We have done our best to address your comments, and hope you find the changes to be positive. Here is a rough summary of overall changes made to the paper:

1) We have included samples from continuous BGAN trained on the full 1000-label 2012 Imagenet dataset and without conditioning on the label (e.g., see "Conditional Image Synthesis With Auxiliary Classifier GANs").
2) We have added a section to the Appendix, 7.1, which quantitatively compares BGAN against WGAN-GP on discrete MNIST and which shows BGAN outperforming WGAN-GP on all metrics estimated, including Wasserstein distance, when comparing the *actual discrete distributions*.
3) We have moved Theorem 2 to the Appendix, Section 7.2, as part of a larger section on the variance-reducing method. In this section is now an experiment that quantitatively validates our variance-reducing method (eq 10 vs eq 8).
5) We have added a paragraph to the related works on the GAN likelihood ratio connection as well as boosted the paragraph on using REINFORCE.
6) We removed the paragraph regarding the connection to classifiers in order to help clean up the paper.
7) We have made various other minor edits following comments from the reviewers.

Thank you,
      - the authors

[Update, Jan 5, 2018]
We have submitted one more revision. We had found a few more minor typos as well as made some additional small alterations to the text. Finally, there were some errors in the values reported in Section 7.1, which have been fixed. This does not change our conclusions in any way.

---

### Decision · Program_Chairs · 2018-01-29
**ICLR 2018 Conference Acceptance Decision**

**Decision:**

Accept (Poster)

**Comment:**

Training GANs to generate discrete data is a hard problem. This paper introduces a principled approach to it that uses importance sampling to estimate the gradient of the generator. The quantitative results, though minimal, appear promising and the generated samples look fine. The writing is clear, if unnecessarily heavy on mathematical notation.